# The Controversial Role of Homocysteine in Neurology: From Labs to Clinical Practice

**DOI:** 10.3390/ijms20010231

**Published:** 2019-01-08

**Authors:** Rita Moretti, Paola Caruso

**Affiliations:** Neurology Clinic, Department of Medical, Surgical, and Health Sciences, University of Trieste, 34149 Trieste, Italy; paolacaruso83@gmail.com

**Keywords:** homocysteine, dementias, stroke, inflammation, vascular endothelium

## Abstract

Homocysteine (Hcy) is a sulfur-containing amino acid that is generated during methionine metabolism. Physiologic Hcy levels are determined primarily by dietary intake and vitamin status. Elevated plasma levels of Hcy can be caused by deficiency of either vitamin B12 or folate. Hyperhomocysteinemia (HHcy) can be responsible of different systemic and neurological disease. Actually, HHcy has been considered as a risk factor for systemic atherosclerosis and cardiovascular disease (CVD) and HHcy has been reported in many neurologic disorders including cognitive impairment and stroke, independent of long-recognized factors such as hyperlipidemia, hypertension, diabetes mellitus, and smoking. HHcy is typically defined as levels >15 micromol/L. Treatment of hyperhomocysteinemia with folic acid and B vitamins seems to be effective in the prevention of the development of atherosclerosis, CVD, and strokes. However, data from literature show controversial results regarding the significance of homocysteine as a risk factor for CVD and stroke and whether patients should be routinely screened for homocysteine. HHcy-induced oxidative stress, endothelial dysfunction, inflammation, smooth muscle cell proliferation, and endoplasmic reticulum (ER) stress have been considered to play an important role in the pathogenesis of several diseases including atherosclerosis and stroke. The aim of our research is to review the possible role of HHcy in neurodegenerative disease and stroke and to understand its pathogenesis.

## 1. Homocysteine Pathways and Regulation 

Homocysteine (Hcy) is a sulfur-containing amino acid related to methionine metabolism [1] and it is either degraded via the remethylation pathway or converted, via the trans-sulfuration pathway, into cysteine (Figure 1). 

The pathway of one-carbon metabolism activates one-carbon units, usually from serine, tightened to tetrahydrofolate. The methionine synthesis is anticipated by the reduction of 5,10-methylenetetrahydrofolate to 5-methyltetrahydrofolate (5-methylTHF) catalyzed by the flavin-containing methylenetetrahydrofolate reductase [2]. 

The produced 5,10-methylenetetrahydrofolate is employed for the production of thymidylate and purines (fundamental for nucleic acid synthesis) and of methionine, fundamental for methylation-process. 5-methyltetrahydrofolate is the substrate to methylate homocysteine, employing vitamin B12 and folate as co-factors.

Methionine adenosyltransferase (MAT) catalyzes S-adenosylmethionine (AdoMet) in a reaction involving methionine and ATP [3,4]. Every reaction made by methyltransferases produces S-adenosylhomocysteine (AdoHcy), which is a potent inhibitor of most of them [5,6]. An AdoHcy hydrolase (SAHH) acts on AdoHcy, producing adenosine and homocysteine, and they need to be metabolized or transported out of the cell to prevent their accumulation [7]. This hydrolysis is a reversible reaction that favors *S*-Adenosyl-l-homocysteine (SAH) synthesis. The S-Adenosyl-Methionine = AdoMet (SAM) to SAH ratio defines the methylation potential of a cell [8]. If homocysteine is allowed to accumulate in normal conditions, it will be rapidly metabolized to SAH, which competes with SAM for the active site on the methyltransferase system [9,10,11,12].

The oxidation of homocysteine to homocysteic acid is one of the potential explanations, with many dangerous effects of homocysteine since it is a mixed excitatory agonist of *N*-methyl-d-aspartate (NMDA) receptors [13]. Homocysteine is also methylated in the entire body, but not in the brain, by betaine [14,15]. Homocysteine remethylation is catalyzed by the methionine synthase (MTR) enzyme, which requires vitamin B12 (Cbl) as a coenzyme [16,17]. 

During the transsulfuration pathway, homocysteine is irreversibly degraded to cysteine. Cysteine is a precursor of glutathione, the most vital endogenous anti-oxidant [18]. In most tissues, homocysteine is either remethylated or exported out of the cell. 

The liver is the main organ of degradation of excess methionine and in maintaining homocysteine at adequate levels [6]. In the liver, there is a univocal correspondence between the increase of methionine and a concomitant increment of AdoMet, and, due to an intrinsic autoregulatory system, AdoMet inhibits methylenetetrahydrofolate reductase (MTHFR) and activates Cystathionine β Synthase (CBS) activity [19,20]. 

Excess of methionine causes an increased homocysteine degradation via the transsulfuration pathway. Lack of methionine conserves homocysteine, via remethylation, back to methionine.

Therefore, 5-methylTHF functions as a methyl donor for homocysteine remethylation [21]. The resulting tetrahydrofolate (THF) can directly be converted into 5,10-methyleneTHF by the action of serine hydroxymethyltransferase (SHMT) [22]. The conversion of THF into 5,10-methyleneTHF is catalyzed by methylenetetrahydrofolate dehydrogenase (MTHFD1) [23]; MTHFR enzyme, therefore, regulates the passage from 5-methyl THF to homocysteine remethylation.

Therefore, to summarize, it is accepted that:Whenever there is a methionine deficit, Hcy can be re-methylated to form methionine, by the employment of N5,N10-methylentetrahydrofolate [24].If there is an adequate amount of methionine, Hcy is employed for the production of cysteine, mediated by cystathionine–beta-synthase, with pyridoxine as a cofactor [25].

## 2. Homcysteine: When Too Much Is Too Much?

The physiologic levels of Hcy in a healthy population are determined primarily by the dietary intakes of methionine [26], folate [27], and vitamin B12 [28].

Recent studies are generally confident with the fact that lifestyle conditions, such as smoking, alcohol consumption, and physical inactivity, may help the elevation of Hcy [29,30].

Normal levels of Hcy range between 5-and 15 micromol/L and in physiological conditions, plasma total (t) Hcy levels are <15 µmol/L, as reported by the majority of investigations. Less frequently, a threshold of 13 µmol/L has been reported, this depending on the method used. Hyperhomocysteinemia (HHcy) is typically defined as levels >15 mol/L in reported studies; levels between 15–30 are considered moderate HHcy; levels at 30–100 micromol/L are considered severe HHCy; levels above 100 micromol/L are considered fatal HHcy [31].

Different epidemiological researches propose that increased homocysteine level is an independent risk factor for vascular diseases including stroke and dementia. HHcy-induced oxidative stress, endothelium dysfunction, inflammation, smooth muscle cell proliferation, and endoplasmic reticulum (ER) stress have been considered to play an important role in the pathogenesis of several diseases including atherosclerosis.

## 3. Homocysteine’s Clinical Role

The clinical role of Hcy, and especially of its accumulation, is frequently controversial in common clinical practice [32]. It seems quite obvious that Hcy is not relevant, per se, at the moment, rather than it is important in clinical practice if accumulated. 

Elevated serum homocysteine relates to an increased risk for vascular disease. The medical interest in this amino acid started in 1969 when a report highlighted that elevated urinary concentrations of Hcy (homocystinuria) in children with inborn errors of Hcy metabolism were associated with vascular damage. In past decades, it has been widely shown that HHcy may cause neurotoxicity. Experimental evidence suggests that elevated plasma homocysteine levels may cause toxicity by a variety of mechanisms, which include direct toxicity and vascular endothelial injury. Elsewhere, it was reported that elevated homocysteine is an independent predictor of poor outcome in patients with stable and acute coronary disease. HHcy is related also with functional disability in the acute phase of stroke. Finally, has been shown that patients with acute stroke with elevated serum homocysteine levels are at an increased risk for early neurological deterioration (END) [33].

Several other factors like age, plasma folate, and vitamin B12 concentrations, serum creatinine, alcohol consumption, dietary restrictions, and different pathological conditions (diabetes, hypertension, renal insufficiency) can be associated to elevated plasma total Hcy levels.

The very first descriptions of damages induced by homocysteine accumulation are quite indirect; they are related to a genetic deficiency of Cystathionine β Synthase (CBS) and to other genetic alterations of remethylation and trans-sulfuration pathways, which induced severe hyper-homocysteinemia (HHCY) (total Hcy > 50 μM) or homocystinuria. Severe HHCY (>100 LM) in children with a CBS defect correlates with a 10-fold increase in concentrations of Hcy in cerebrospinal fluid (CSF) [34]. It happens that, when there is an MTR deficiency or dysfunction, 5-methylTHF cannot cycle and the entire process produces an accumulation of 5-methylTHF; folate does not circulate, limiting the synthesis of purines and thymidine, due to a severe alteration and inhibition of the trans-metilation pathway. Hcy accumulation leads to a delay or even an abolition of the closure process of the neural tube [35]. 

Moreover, the methylation reactions are strongly necessary in the brain for the fact that SAM is the sole donor in numerous methylation reactions involving proteins, phospholipids, and biogenic amines [23], and for packaging of many phospholipids [31]. This way, alterations of methylation with consequent Hcy accumulation are strong determinants for many congenital neural tube and central nervous system alterations [36].

To be precise, the trans-sulfuration pathway, fundamental, as sees above, for Hcy catabolism is quite enigmatic in the brain: the most studied site of trans-sulfuration is the liver, where it is responsible for the glutathione synthesis; any specific data has been reported for the brain. Nevertheless, we testify that many works suggested that, unexpectedly, cystathionine beta-synthase and cystathionase, act in the brain [37,38], and could promote the trans-sulfuration of Hcy into cysteine, producing, therefore, the precursor of glutathione [39]. A single paper documented an in vitro model production of glutathione by astroglia by cysteine and cystathionine [40]. 

It should be, therefore, reconsidered the trans-sulfuration cerebral pathway for glutathione synthesis in the brain [41,42]. Being that cysteine can be considered the rate-limiting substrate for the synthesis of glutathione, it has been labelled a Na(Sodium)-dependent glutamate transporter, which vehicles cysteine in astrocytes. Different cysteine precursors, present in the brain (cystathionine, homocysteine, and methionine), could be considered as the effective originators of glutathione in the brain [43]. 

A saturable Hcy receptor has been detected in animal models and it works by facilitating a diffusion process of Hcy inside the brain [44], which, by the way, is the only known system of Hcy transport. Moreover, although an intrinsic cerebral production of Hcy in human brain has been documented, probably showing different regional variability, it has not been further investigated [45,46].

## 4. Folic Acid, Vitamin B12 and Hcy: Their Relationships

There is an intimate relationship between folic acid and vitamin B12 [47]. Treating a B12 deficient patient with folate or conversely a folate deficient patient with B12 may exacerbate the neurologic consequences or either deficiency [48,49,50]. 

Avoiding treating a B12 deficient patient with folate, which might exacerbate the neurological consequences of either deficiency, the good clinical practice recommends that cyanocobalamin deficiencies should be excluded before folate supplementation is commenced, or if necessary, it should be appropriate to supplement folate and vitamin B12 together [51].

The National Institutes of Health declared that supplementation of large amounts of folic acid can mask the damaging effects of vitamin B12 deficiency [52]. This problem can be determined by masking effects of laboratory evidence of megaloblastic anemia, originally caused by vitamin B12 deficiency [53]. It has been argued that where there is a defect in homocysteine methyltrasnferase, or when there is a combined deficiency of B12 and Hcy-methyltransferase, a specific reaction happens, called methyl-trap of tetrahydrofolate (THF); THF is converted to a reservoir of methyl-THF and therefore, folic acid is trapped and cannot be employed anymore [54].

Hcy is a sulfur-containing aminoacid, tightly related to methionine metabolism [23] the causative factors of accumulation of Hcy can be different due to different genetic pathway defects, or to mutation of enzymatic cascade or to the defects of vitamin B12 and folate, during human life [1].

In effect, an increase of Hcy occurs in the brain and CSF, and in the plasma, within the aging process and inside several neurological diseases [55,56].

It has been demonstrated in animal models that Hcy could be intrinsically toxic, compromising the integrity of the blood-brain barrier [57]; the same mechanism has been postulated for human beings [58,59]. Neurological damages have been reported in mice deficient in CBS enzyme (CBS −/+ or CBS −/−), where Hcy increased by approximately 2–50-fold in comparison to wild-type mice [60]. 

The most frequent causes of HHcy in adult life are the genetic enzyme deficits involved in Hcy metabolism, mainly MTHFR, methionine synthase and CBS [61] and from nutritional deficiencies of folate, vitamin B6 and B12. The most frequent form of is a point-mutation (C-t substitution at nucleotide 677) in the MTHFR, which is associated with a thermo labile variant of activity of the enzyme, that has half-normal activity [62,63]. As previously reported, HHcy can derive from vitamin B6 and B12 deficiency and folate deprivation. In general, vitamin levels are related inversely to Hcy; therefore, their lack determine HHcy.

As well reported by Ganguly and Alam [24], being that SAM-to SAH ratio is the expression of the methylation potential of a cell, “HHcy tends to decrease the methylation potential”. Therefore, Hcy can induce a global DNA hypomethylation and suppress the transcription of cyclin A in endothelial cells; on te contrary, Hcy leads also to up-regulation, by the same hypomethylation, of some other genes, causing an increase of *p66shc* expression in endothelial cells, thus contributing to oxidant stress [8,24].

The mechanisms of damage promoted by Hcy are various, acting as a promoter of neurodegeneration, or inflammation, finally inducing also cerebrovascular diseases.

## 5. Mechanisms of Damage Induced by Hcy

### 5.1. Homocysteine and Neurodegeneration

A classic experiment made to discover the direct effect of Hcy in the brain has been made by direct application of Hcy, by two different drug-delivery methods, pressure ejection and ionophoresis [64]. Both the ways of delivery produce an evident increase of D,L-Hcy and L-Glutamate, implying a possible direct excitatory action of it on neurons.

The mechanism of damage evoked for Hcy excitatory role has been found [65,66]: Hcy is an agonist of the endogenous glutamate receptors, NMDA receptors [67,68].

Through Hcy-NMDA binding, Hcy indirectly enhances calcium influx [69]. This is not a constant reaction, and it largely depends in the Glycine concentration; when glycine is in normal concentration (10 μmol/L), Hcy acts as a partial antagonist of the glycine site of the NMDA receptor, and it inhibits the receptor-mediated activity, acting as a neuroprotective factor [23,65]. Therefore, it can be easily demonstrated that when glycine levels are normal, only HHcy could exert a toxic effect (i.e., Hcy = 100 μmol/L). 

On the contrary, when glycine levels are higher inside the brain, more than 10 μmol/L, (and this occurs in clinical conditions in different scenario: brain ischemia, head trauma, or even protracted migraine cluster), even a low concentration of Hcy (i.e., Hcy = 10 μmol/L) could be an agonist on NMDA [70,71], exerting an excitatory action, and enhancing calcium influx. 

More recent data underlies a new possible mechanism of Hcy’s action: its direct activation of the group I metabotropic glutamate receptors, by competing with inhibitory neurotransmitters, such as GABA [70], inducing also this way an increase of calcium influx. 

Many clinical works try to focus the possible direct consequences of Hcy inside neurodegenerative disorders: it is well-accepted that Hcy increases in CSF with ageing. Moreover, some works show a direct correlation between Hcy increase and Abeta 1–40 deposition in the brain of AD patients [72]. It seems that Hcy can induce and even potentiate the intracellular and extracellular accumulation of Abeta 42 [73], amplifying even the harm effects derived by Abeta 42 deposition [74,75]. Hcy increases the toxicity of Abeta on the vascular smooth muscle cells of small brain arteries [76]. It has been documented, in fact, that an endoplasmic protein-Hcy related (HERP) in the presence of Hcy potentiates the c-secretase enzyme activity, promoting a major Ab1-40 accumulation inside the brain [77]. Soluble oligomers of amyloid beta could change the redox state with DNA methylation and gene transcription inhibiting a transporter 3-EAAT3-mediated cysteine uptake and lead to HHcy [78]. HHcy, by DNA hypomethylation, as above reported, can lead to up-regulation of presenilin genes, in particular, the one regulating presenilin 1 (PS1). PS1 is tightly related to methylation process in the brain, but above all, it promotes the amyloid precursor protein (APP) synthesis [79,80]. The HHcy induction up-regulates PS1 gene, and therefore increases APP, promoting, therefore, the amyloid cascade sequence. 

Another protein is directly involved in many neurodegenerative pathologies is the tau protein: it seems to act as a coordinator of the assembly of microtubules, permitting a correct axonal transport. 

The protein phosphatase methyltransferase 1 (PPM1), whose methylation is SAM-dependent, regulates the activity of the protein phosphatase methyltransferase 2A (PP2A), which acts as a dephosphorylating system for tau protein [81,82,83,84,85]. 

Tau hyper-phosphorylation inhibits the congregation of microtubule; their precipitation determines the deposition of the neurofibrillary tangles. Hence, the reduced methylation capacity increases the hyperphosphorilated-tau (P-TAU).

It has been documented a post-translational modification of PP2A stability in AD patients, [86], which can be related to lower levels of SAM (or to an increase of SAH, for the SAM-to SAH ratio, above described) [87,88], implying a potential increase of P-TAU, with the consequent neurofibrillary depositions. The induced depletion of folic acid in neuroblastoma cultured cells, causing therefore the most common HHCY condition produced an increase of P-TAU by 66% [46].

Moreover, it seems quite interesting that Hcy leads to an induction of m-RNA and protein expression of a specific protein, C-reactive protein (CRP), augmenting the NR1 subunit of NMDA receptor expression, HCy can promote a pro-inflammatory response in vascular smooth muscle cells of small brain arteries, by stimulating CRP production, usually enhanced by a combined NMDA-ROS-erk1/2/p38-nfKBeta signal pathway [89]. Not only, this way Hcy might be a promoter of atherosclerosis system, but also, small arteries can promote neurodegeneration, diminishing other capabilities of autoregulation, due to their role in autoregulation, leading to an alteration of the blood-brain barrier. This way, Hcy might potentiate its direct neurotoxic effects [90].

Moreover, HHcy accelerates the dopaminergic cell death, probably due to the fact that HHcy could cause a severe reduction in dopamine turnover in the striatum [91]. It has been suggested that there is an ARG-rich domain, which is located in the middle portion of the third loop of the D2 receptor, which has high affinity for Hcy. Hcy seems to have an allosteric antagonist activity of D2 receptors [92].

### 5.2. Hcy, ROS, Inflammation

Recently, a well-conducted study by Curro et al. (2014) [61] conducted on neuroblastoma cells incubated with Hcy determined some different and time- and concentration-dependent results. 80 microM Hcy exposure produced 80% of cell death after 5 days of incubation; 40 microM Hcy conducted to a 35% of cell death after 5 days of incubation; quite interesting, cell exposure to Hcy for three days does not induce any change in Reactive Oxygen Species (ROS), but exposure to Hcy for five days elevated to a 4.4 fold increase ROS production; a five days incubation with Hcy induced a 2-fold increase of bax mRNA and of 14-fold of Bcl-2 mRNA; a three-days incubation with Hcy induces an increase of 2-fold for cyclin D1 mRNA, 6-fold for cyclin E1 mRNA and 5-fold for cyclin A1 mRNA. Unexpectedly, all the levels turn back to a normal range after 5 days incubation. 

What this study points out is that there is a general upregulation of p21 and p-16 after 5 days of Hcy incubation, inducing a reduction of 35% of pRB, checkpoint regulators of G1 cell-cycle phase. This work suggests a potential genotoxic stress induced by Hcy exposure. In response to the high Hcy level, endothelial cells produce NO to induce the formation of S-nitrose-Hcy, which acts as a protector of endothelium; however, with chronic exposure to Hcy, NO levels diminish [93] and this fact, associated to the high levels of Hcy, promotes endothelial damage. The first by stimulation of muscle cells, vasoconstriction and promoting inflammatory response, testified by an increase of c-reactive protein and cysteinyl leukotrienes, was associated with an incremental increase in HMG-CoA reductase activity [94]. 

The activities of methionine synthase that mediate the clearance of Hcy is linked to the redox potential of the cells [95,96], with an observed efficacy in oxidative stress process; in this situation, more Hcy is converted into cysteine and glutathione. A disruption of the CBS causes altered redox homeostasis, and through a reduction of the cysteine and glutathione, it causes an alteration of oxidative repairing process [97]. The endothelial damage is mediated by one of the precursors, hydrogen sulfide (H2S), which is formed during the transsulfuration process [98,99]. 

The disruption of the redox system in vascular and neuronal cells [100] induces and accelerates the lipid peroxidation sequel of events [101,102]. The vascular endothelium is a single layer of dynamic cells which, through a variety of stimuli, produces vasoactive substances to maintain vascular tone and regulates blood flow to the tissues, and among these, this effect was attributed to a substance(s), subsequently identified as nitric oxide (NO) [103].

Endothelial dysfunction results from a disruption in the cellular integrity, leading to impaired endothelium-dependent relaxation mainly due to a reduction in the NO bioavailability. NO is produced from its precursor L-arginine by endothelial nitric oxide synthase (eNOS). Under physiological conditions, following production, NO diffuses across the endothelial cell membrane into the vascular smooth muscle cells to activate guanylate cyclase, leading to subsequent cyclic guanosine-3′,5-monophosphate (cGMP)-mediated vasodilation. Several molecules such as acetylcholine, bradykinin, serotonin, and substance P can induce eNOS. Another important stimulus is the shear stress exerted by the flowing blood which can cause ion channel activation for a rapid response or through a process of phosphorylation induce sustained release of NO to maintain vasodilation [104].

HHcy-induced ROS production decreases NO production and bioavailability triggering increased redox signaling. Impaired NO production during HHcy can also occur due to inhibition of Dimethylarginine dimethylaminohydrolase (DDAH) causing Asymmetric dimethylarginine (ADMA) accumulation [105,106,107,108].

Therefore, Hcy has been linked to an increment of ROS and deactivation of nitric oxide, with the well-known inflammation cascade [101,102]. 

Another possible link has been reported between Hcy and lowered melatonin production [109]. It has been demonstrated that melatonin scavenges free radicals [110] and counteracts Hcy by a direct antioxidant effect and by apoptosis modulation [111,112].

Different studies demonstrate that the antioxidants, such as N-acetyl cysteine, vitamin E or C might reduce the potential pro-inflammatory response of Hcy in animal models [113,114]. 

Different in vivo reports recognized that the Th1-activity induced the Hcy inflammation response [115], and it appears that HHcy can be detected in chronic inflammatory conditions, even if vitamin B12 and folate are in range. 

In a double-blind interventional study, though, the logical implementation with vitamin B complex does not affect inflammatory markers, such as neopterin, Il-6, CRP, etc.; on the contrary, the implementation of folate reduced neopterin, suggesting a possible modulating role of folic acid in the inflammatory cascade [116]. 

Interestingly, multiple traumatism and secondary septic status associated with a systemic inflammatory response has been associated to HHcy, and the constancy of this report is related to a poor clinical outcome [117]. Unusually, these patients did not show any other factorial causes (such as folate or B12 poorness). The authors hypothesized that the pro-inflammatory condition of these patients leads to strong activation of macrophage-system cascade by Hcy, with a consequent release of ample amounts of ROS, potentiating the oxidative stress.

These results have been supported by a definite activation of B lymphocyte-induced by Hcy; this process seems to determine an increase of pyruvate kinase muscle isozyme 2 (PKM-2) in B cells. Its inhibition, employing shikonin, causes the restore of the metabolic changes induced by Hcy. PKM-2 seems to suggest the so-called metabolic accelerated initiation of atherosclerosis cascade mediated by HHcy, in vivo and in vitro [118].

Li et al. [119] showed that in animal models, an induced hyper-Hcy produces a higher plasma level of TNF-α and IL-1beta, and an apparent decrease of plasma levels of H2S and cystathionine gamma-lyase expression in the peritoneal macrophages. It has been demonstrated that hyper-Hcy inhibits cystathionine γ-lyase expression and H2S production in macrophages; HHcy is related to an increase of the DNA expression of methyltransferase and hypermethylation process in promoter regions, therefore inducing a promoter trigger of inflammation [120,121]. Li et al. [119] definitively demonstrated that cultured macrophages cells exposed to Hcy showed a memory response, probably induced by epigenetic mutations, that influences the expression of promoter genes inflammatory response and endothelium atherogenesis. A single study demonstrated an in-vitro Hcy alteration of the transcriptional repression of fibroblast growth factor 2 [122]. 

There is a possible link between the excitotoxic effect of Hcy and the pro-inflammatory role of Hcy: the NMDA receptors are found not only in neurons (see above in the text), but also on neutrophils and macrophages. The activation of these receptors, as well as in the cerebral context, arises the cytoplasmatic calcium influx, and activates a pro-inflammatory cascade, with an accumulation of ROS species [123]. In fact, in the Rheumatoid Arthritis, as an example, [124] HHcy is two times more frequent than in the general population and HHcy contributes to the oxidative stress, and, indirectly, by the excess of ROS released, it induces an up-regulation of the Nuclear Factor Kappa B, considered as one of “the master regulator of the expression of inflammatory genes” [125].

It is widely accepted that there is a direct correlation between Hcy and SAH, in the reaction mediated by SAH-hydrolase; therefore, a higher level of SA reduces the SAM concentration and even the higher SAH level (or, the lower SAM level) induces an oxidative reaction and might explain Hcy neurotoxicity [126]. Increasing SAM levels, or employing enriched SAM pabulum for cells neuronal tissues, might reduce apoptosis by 50% [127,128]. 

On the other side, asymmetric dimethylarginine (ADMA) is an endogenous inhibitor of endogenous nitric oxide synthase (eNOS), the enzyme catalyzing the formation of nitric oxide (NO) from arginine. Similarly to Hcy, ADMA also represents an important factor correlating with endothelial dysfunction. HHcy may also be associated with reduced plasma levels of NO and impaired endothelium-dependent vasodilation. Plasma levels of ADMA have been reported to be positively correlated with plasma homocysteine levels. Lowering levels of Hcy and ADMA seems to reduce the progression of atherosclerosis and prevent atherothrombotic diseases [129,130].

### 5.3. Hyperhomocysteinemia and Cerebrovascular Disease

Development of hyperhomocysteinemia is a characteristic feature of aging. HHcy contributes to the development of age-associated disorders, like endothelial dysfunction, decline in renal and cognitive functioning. Increased levels of Hcy are seen in 5–7% of the general population [131].

Increasing evidence showed that homocysteine is associated with different kind of cardiovascular and cerebrovascular diseases [132,133]. 

A positive correlation between Hcy levels and ischemic heart disease has been widely reported, higher level of Hcy (>15 μmol/L) is considered a risk factor in the development of cardiovascular diseases and ischemic heart disease. The subtended mechanism may involve endothelial injury and promoted platelet accumulation at the site of endothelial injury, primarily through oxidation type reactions. Reduction in Hcy concentration lowers the risk of ischemic heart disease by one-third [134,135].

Homocysteinemia is a contributing factor for thrombosis and can be combined with other thrombophilic factors in thrombophilia. Homocysteinemia can be seen secondarily in folate and cobalamine deficiencies. Vitamin B12 is the co-enzyme for methyl donation from 5-methyltethrahydrofolate in tetrahydrofolate, necessary for methionine synthetase and helps in the conversion of homocysteine to methionine and the changeover of methylmalonyl coenzyme A to succinyl coenzyme A. Folate promotes the remethylation of homocysteine which can induce DNA strand breakage, oxidative stress, and apoptosis. Vitamin B12 and folate actively promote acid synthesis and the methylation reactions, and their lack causes the inhibition of S-adenosylmethionine, an accumulation of homocysteine, which causes direct damage to the vascular endothelium and inhibition of *N*-methyl-d-Aspartate receptors [136,137,138,139]. HHcy represents a potentially modifiable risk factor for stroke. Increased levels of Hcy may cause neurotoxicity and inflammatory activity and affect coagulation by the role of tissue factor expression. Direct toxicity and vascular endothelial injury in ischemic stroke patients whose inflammatory system reacts most intensely may be at greater risk for cardiovascular and cerebrovascular diseases.

It has been reported that HHcy relates with both stroke (ischemic or hemorrhagic, via inducing cerebrovascular atherosclerosis and atherothrombosis in the first case, and probably via upregulation of matrix metalloproteinases-9 (MMP-9) expression, which takes responsibility for atherosclerotic plaque instability and rupture, in the second ones [140]) and intracranial hemorrhage incidence.

Moreover, HHcy is an independent predictor of poor outcome in patients with stable and acute coronary diseases. Finally, patients with stroke and HHcy present an increased risk for early neurological deterioration, long-term mortality, and poor function outcome. HHcy is an independent risk factor for severity of stroke and coronary heart disease [33,141,142,143]. 

An Italian study conducted in elderly patients with acute ischemic stroke admitted to a Stroke Unit (SU) reported that high admission plasma total Hcy was unrelated to mortality during recovery but was linked with poor functional status at discharge for those discharged alive [144]. 

Similar data have been reported elsewhere. Higher homocysteine levels are associated with early neurological deterioration (END) in acute ischemic stroke, defined END as an incremental increase in the NIHSS score by ≥1 point in motor power, or ≥2 points in the total score within the first week after admission. Moreover, the risk of early neurological deterioration rises with increasing Hcy levels, worsening the already high stroke burden for the society [33].

Zhihong Shi et al. in a Chinese cohort demonstrate that elevated total Hcy levels during the acute phase of an ischemic stroke significantly predict mortality with a higher increased risk of death 48 months after stroke. Moreover, was seen that the association between total Hcy levels and all-cause mortality was only significant in patients with the large-artery atherosclerosis stroke subtype, while the association was not significant in the small-vessel stroke subtype. Suggesting that either elevated Hcy levels in the acute phase of a stroke may be more detrimental in large-vessel strokes compared with small-vessel strokes or that Hcy levels increase in large vessel strokes but remain unchanging in serious and minor small-vessel strokes [142]. 

Has also been reported that patients with acute ischemic stroke have high levels of the endogenous NOS inhibitor ADMA. Since Hcy levels were positively correlated with ADMA levels, ADMA may be a mediator of the atherogenic effects of Hcy. In those patients have been reported that supplementation with folic acid and vitamin B12 decreased plasma levels of ADMA and Hcy, proposing the association of ADMA-mediated mechanisms in the decrease of NO activity in these patients [145].

Anyway, data from literature are thus controversial. The relationship between HHcy levels and stroke recurrence and overall cardiovascular events after index stroke remains debatable [146,147]. 

In their work Perini et al. found that Hcy in the acute phase of stroke was not associated with stroke severity or outcome. They confirmed that elevated Hcy levels were associated with both ischaemic and haemorrhagic stroke [148]. Similarly, Haapaniemi et al. doubt correlation between Hcy and stroke suggesting that Hcy levels in patients were significantly lower on admission but not at later time points, with levels increasing by week and remaining at this level for longer period. Similar considerations were reported also elsewhere [149].

Concerning stroke treatment and prevention has been written that some therapies such antiplatelet cure (aspirin, clopidogrel) and statins, may decrease inflammatory mediator levels [150]; but the effect of these drugs on high serum Hcy levels and stroke or cardiovascular events outcomes remains unknown.

Elevated plasma Hcy levels relate also with intracerebral hemorrhage (ICH). Particularly has been reported a strictly link between Hcy levels and larger hematoma volume, an independent predictor of mortality and poor neurologic outcome after acute ICH. Hcy levels might ruin vessel wall integrity and disrupt cerebrovascular permeability through endothelial dysfunction, elastic structures damage, and basal lamina injury. Fangfang Zhou et al. reported an association between Hcy levels and hematoma volumes in Thalamoganglionic ICH, but not in patients with lobar or infratentorial ICH. Finally, the Hcy levels seem to be unrelated with 6-month clinical outcome [151]. Generally, an association with higher Hcy levels and increasing CMBs was detected, while no association between Hcy level and the severity of the CMBs was seen.

Higher Hcy value can be seen also in other different cerebrovascular diseases like Cerebral venous thrombosis (CVT). In CVT deficiencies of protein C, protein S, antithrombin III, factor V Leiden mutation and hyperhomocysteinemia is common [152].

Finally, Hongzhi Luo et al. found that HHcy may have a great impact on the pathogenesis of Cervical artery dissection (CAD). The photogenic mechanism remains unknown, but on the basis remains that the consideration that HHcy promotes oxidative stress, accelerates vascular smooth muscle cell migration, adventitial collagen accumulation, and neointima formation. Moreover, within different risk factors for CAD Methylenetetrahydrofolate reductase (MTHFR) is a rate-limiting enzyme in the methionine metabolism pathway that catalyzes the conversion of 5,10-methylenetetrahydrofolate to 5-methyltetrahydrofolate, a cosubstrate for homocysteine remethylation to methionine. Individuals with the 677TT mutation of MTHFR are predisposed to HHcy that may lead to CAD incidence. Lowering the concentration of homocysteine could be a promising tool in CAD intervention [153].

## 6. Hcy in Real Clinical Practice

Clinical trials and studies failed to demonstrate clear and definite results, considering the supplementation in patients or healthy population of vitamin B12, folate or both, either considering HHcy either in not evidence augmented levels of Hcy. Many criticisms may be lead towards the trials implemented, well summarized by Price et al., 2018 [31].

Many studies evidenced lower serum vitamin B12 levels in subjects with Alzheimer disease (AD) or other dementias [154,155]. 

One of the most profound studies on the topic examined the relationship between vitamin B12 serum levels and cognitive and neuropsychiatric symptoms in dementia, but results seem inconclusive [156]. There are many different studies, which have documented the effectiveness of vitamin B12 supplementation, in improving cognition in demented patients [157,158,159].

Another study demonstrated that vitamin B12 treatment might improve frontal lobe and language function in patients with cognitive impairment, but rarely reverses dementia [160].

Many other works have failed to confirm the optimistic results [161,162], even declaring that subnormal serum vitamin B12 levels are not an important cause of reversible dementia. 

When taken into account the works, which considered the role of the combined low serum levels of vitamin B12 and folic acid [163], it can be evicted that in old people who are not demented, neither low level of vitamin B12 nor folate alone significantly affected the risk of developing AD. On the contrary, it can be observed that, compared with subjects with normal levels of vitamin B12 and folate, the very old patients with the lack of these vitamins showed high risk of developing AD [138]. Hassing et al. [164], showed that AD patients, correctly supplied with vitamin B12 and folic acid did not improve, but observed that they performed better in neuropsychological test. Another study on this topic, the so-called SENECA (longitudinal, 5-years lasting, multi-center study) [165], showed no correlation between mental health and low levels of vitamin B12/folate status.

Concerning purposely the Hcy level, in order to supplement, there are some studies, like the one by Nilsson et al. [166], which evidenced that patients affected by AD, who had elevated homocysteine plasmatic levels, and have been supplied by vitamin B12, perform better in neuropsychological tests; on the contrary, if dementia’s scores are worse, no significant effect emerged from vitamin B12 supplementation. After this study, Bryan et al. [167], obtained a significant improvement in memory recall, speediness of thought, of executive functions, and linguistic appropriateness, and any effect on mood, when more than 200 healthy middle-aged or older 

More recently [168], a large international double-blind placebo-controlled randomized trial, focused on many thousand women, who have previously experienced neural tube defect (NTD) pregnancy; these women have been supplied by large doses of folic acid, and a reduction of NTD of 72% in the following pregnancy was demonstrated [169]. In 1998, the obligatory folic acid fortification of cereal grain products had a direct consequence of an NTD decline rate of 20%. Data have been recently reproduced in larger studies [170]. Several mechanisms have been suggested to explain the possible link between folic acid lack and NTD [6]. As above reported, folic acid is related to DNA methylation, a fundamental regulating system in precocious embryogenesis [171,172]. Folic acid relays on the correct functioning of the MTHFR enzyme; MTHFR acts as a donor of methyl groups, necessary for purines and pyrimidine synthesis. Therefore, genetic micro-alterations of MTHFR (for example 677TT genotype) decrease the methylation reactions, with a consequent reduction of global DNA methylation, related to NTD. NTD has been demonstrated in vitro, by the inactivation of DNA methyltransferase (DNMT3B) [173,174,175,176].

A reduced AdoMet/AdoHcy ratio, which causes an increase of Ado Hcy, and therefore inhibits DNA methylation has been found also in a single case of reported co-existence of trisomy 21 and spina bifida [177]. Folic acid level and homocysteine are indirectly related to years of age, they have been related to aging process [178,179], and, therefore, to cerebrovascular pathologies, whose main un-modifiable risk factor remains aging process. The Canadian Study of Health and Ageing [180], a 5 years population-based survey, studied the potential risk of any cerebrovascular event, including vascular dementia, vascular cognitive impairment or fatal stroke, during a strict follow-up of serum folate levels. After adjusting for normal vascular risk factors (such as smoking, nutrition, diabetes, etc), it emerged that the risk for a vascular event, associated with the lowest folate quartile was OR 2.42 (95% CI; 1.04–5.61). The same study demonstrated that a low folate level represents a higher risk for the development of vascular events, even in female patients (OR 4.02, 95% CI; 1.37–11.81). In the Kingsholmen aging and dementia project, in 250 old (75 to 96 years) and in 71 very old (90 to 101 years), otherwise healthy subjects, the altered memory recall has been related to folic acid low levels [181]. These findings have been reproduced with a significant correlation between lower cognitive function and raised homocysteine level and low folic acid levels [182].

In Table 1, we report the correlation between HHcy and main neurological conditions in accordance with the cited studies from literature.

## 7. Limitations and Further Possibilities

Even with very high knowledge of the biochemical properties and profile of Hcy, lab results are more encouraging than clinical trials, and although combined folic and vitamin B12 therapy substantially reduces homocysteine accumulation, results from randomized placebo-controlled clinical trials have fallen short of expectations [200].

Many are the factors that limit the clinical applicability (homogeneity of recruitment, time of onset, age of onset of HHcy and of the pathology, eventual comorbidities, etc.); duration of the intervention must also be considered, given i.e., that normal cognition generally decline only by-0.1 points on MMSE each year [201]. And study designs should take into consideration even the other B vitamin status (such as B1 and B6, [31]). However, as Nichols recently pointed out [202], blood sample for Hcy should be taken and lowering it with vitamin implementation should be done, due to low cost and fast application. However, it is mandatory that we need more evidence on doing this as a primary preventive strategy “to clinch the argument”.

## 8. Conclusions

The role of HHcy in several neurological and cardiovascular diseases is still unclear. HHcy seems to be an independent risk factor for cerebrovascular disease, dementia, and many other disorders, and, moreover, is related with atherosclerosis and cardiovascular disease. It has been postulated that Hcy has a causative role in the determination of neurological damages due to its neurotoxic effect and to its direct or indirect vascular and endothelium induced pro-inflammatory effect. Anyway, in literature, there are controversial opinions. The efficacy of combined folic acid, B6, and B12-vitamin supplementation to reduce HHcy is well documented but is sometimes hard to find who can really benefits from it.

In cognitive impairment and dementia, Hcy might act to potentiate the effects of Abeta deposition, augmenting its toxic effects by modifying presenilin functions and occasionally interfering with hyperphosphorylation of tau protein. Moreover, Abeta deposition might induce damage to smooth muscle vascular structures and can promote caspase activity. The association between hyper-HCY and worsening of cognitive performance, Amyloid burden, and white matter hyperintensities, has been confirmed in the majority the clinical studies considered. 

HHcy relates with both stroke, ischemic or hemorrhagic, and intracranial hemorrhage incidence, independent of long-recognized factors such as hyperlipidemia, hypertension, diabetes mellitus, and smoking.

Hcy remains, at the moment, a biological marker, with all its limits and without the key of solution for all the neuronal pathologies; EFSA [203] maintains a cautious position: it clearly stated that the maintenance of normal Hcy metabolism has a beneficial physiological effect. All the other protective, antioxidant, and anti-inflammatory effects should be proven.

Therefore, the precise role of HHcy in the development of stroke and dementia should be further studied to dissect out the complex mechanism and evaluate on larger scale trials.

## Figures and Tables

**Figure 1 ijms-20-00231-f001:**
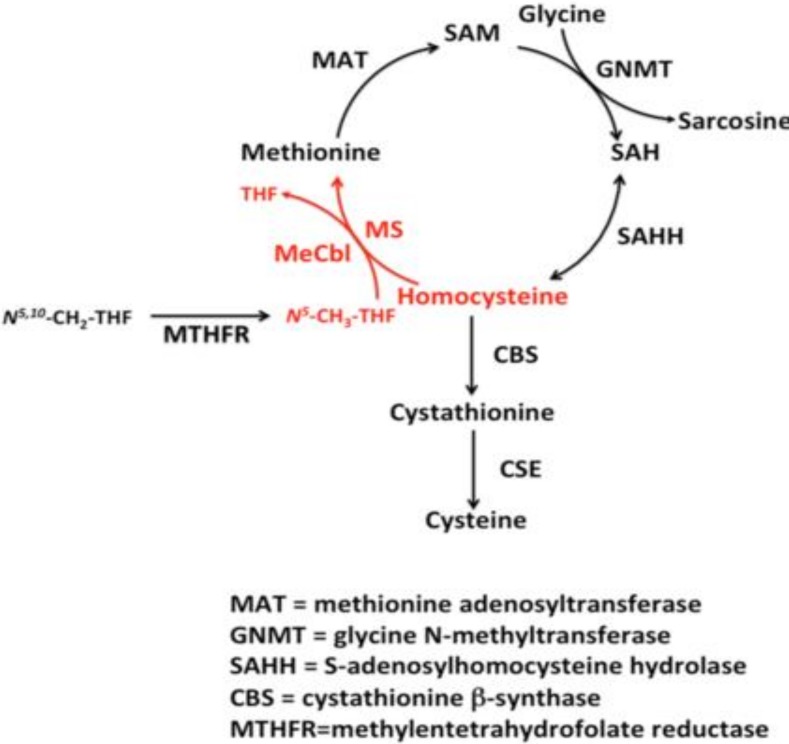
Homocysteine (Hcy) Metabolism.

**Table 1 ijms-20-00231-t001:** Hhcy and main neurological conditions.

Hhcy and Neurological Conditions	Studies	Results
Stroke	HHcy preclinical marker of stroke [183]	Confirmative
	HHcy and prothrombotic [184]	Confirmative
	HHcy and platelet peroxidation [185]	Confirmative
	HHcy and increased pulsatility index in intracranial arteries [186]	Confirmative
	HHcy relationship with the progression of aortic atheroma [187]	Confirmative
Mild cognitive impairment (MCI)	Hhcy as a marker of transition from MCI to dementia [188]	Confirmative
	HHcy correlates with hippocampal function [189]	Confirmative
	HHcy correlates with atrophy progression [190]	Confirmative
	HHCY correlates with the passage from healthy brain to MCI [191]	Not-confirmative
AD	HHcy correlated with AD diagnosis [192,193]	Confirmative
	HHcy correlates with temporal atrophy progression [194]	Confirmative
	HHcy correlate with AD diagnosis [195]	Non-confirmative
PD	HHCy involved in PD pathogenesis [196]	Confirmative
	HHcy involved in augmentation of dopaminergic susceptibility [197,198]	Confirmative
	HHcy induced by Levo-Dopa treatment [199]	Confirmative

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
