# Peer review of "The Controversial Role of Homocysteine in Neurology: From Labs to Clinical Practice"

_ijms, 2019, doi:10.3390/ijms20010231_

Round 1
Reviewer 1 Report
Title not expressing the merit of the paper
Recent publication, comments and novel insights are not fully addressed, newer information from all the neuroscientific branches not included, clinical approach is expressed only marginaly
Stratification of patients with particular neurological disorders- absent
Model studies on particular disoders not fully presented
Molecular mechanism which can elucidate and explain controversies or dicrepancies in the role of Hcy are not fully addressed
If this would be a positional paper for clinician- too superficial, no table, figures
Dietary approach and comments from this point are obvious, if it would be oriented clinicaly, it needs deeper elaboration and critical discussion with recent knowledge
Author Response
Answers to reviewer’s comments
We would like to thank you for the exhaustive revisions and for the opportunity you have given us.
We have managed with the editorial corrections.
Reviewer 1
· Title not expressing the merit of the paper
· Recent publication, comments and novel insights are not fully addressed, newer information from all the neuroscientific branches not included, clinical approach is expressed only marginaly
· Stratification of patients with particular neurological disorders- absent
· Model studies on particular disoders not fully presented
· Molecular mechanism which can elucidate and explain controversies or dicrepancies in the role of Hcy are not fully addressed
· If this would be a positional paper for clinician- too superficial, no table, figures
· Dietary approach and comments from this point are obvious, if it would be oriented clinicaly, it needs deeper elaboration and critical discussion with recent knowledge
Reviewer 1 Answers
· Thank you Sir, we provided to modify the title of our manuscript;
· Dear sir, thank you for your revision, which helps us and gives us the opportunity to revise the ms, inserting more data from literature, and exposing better the clinical approach;
· Dear sir, we provided to report stratification of patients with neurological disorders and HHcy, adding description of model studies;
· Dear sir, molecular mechanism which can elucidate and explain the role of Hcy have been reported;
· Dear sir, we examine in depth the ms, we included table and figure;
· Dear sir, thank you for your revision, deeper elaboration and critical discussion of dietary approach and its clinical role, have been reported.
Reviewer 2 Report
The authors try to discuss the role the controversial role of homocysteine in neurological practice. The review is quite interesting and the authors have made a great effort to collect all important information about homocysteine (Hcy) but the way the whole information is presented causes a great confusion to the reader. The pathways in which Hcy is involved are difficult to understand without a scheme for a scientist not dealing with the same topic. Furthermore the conection of the Hcy levels and a disease or a state is influenced by so many other factors that some times the end point is not clear.
Thus the authors should
1.rearrange the information by separating the major paragraphs in smaller ones with subtitles i.e .1.1, 1.2.
2, present the major pathways in whichHcy involved in order to make easier for the reader to understand the role of SAM, or THF etc Clear schemes that depict the relative mechanisms are very helpful.
3. make approriate tables reporting possible associations between Hcy and a state or disease and the involved factors mentioned in the text. For example Stroke , hcy levels, outcome or effect on disease evolution, etc
4.Correct English language
Minor comment
Although the authors mention the role of hcy in cardiovascular diseases as a main point in their conclusion this information does not make separate part of their text, thus they should rearrange the beginning of the abstract and the conclusion
Author Response
Answers to reviewer’s comments
We would like to thank you for the exhaustive revisions and for the opportunity you have given us.
We have managed with the editorial corrections.
Reviewer 2
The authors try to discuss the role the controversial role of homocysteine in neurological practice. The review is quite interesting and the authors have made a great effort to collect all important information about homocysteine (Hcy) but the way the whole information is presented causes a great confusion to the reader. The pathways in which Hcy is involved are difficult to understand without a scheme for a scientist not dealing with the same topic. Furthermore the conection of the Hcy levels and a disease or a state is influenced by so many other factors that some times the end point is not clear.
Thus the authors should
1.rearrange the information by separating the major paragraphs in smaller ones with subtitles i.e .1.1, 1.2.
2, present the major pathways in whichHcy involved in order to make easier for the reader to understand the role of SAM, or THF etc Clear schemes that depict the relative mechanisms are very helpful.
3. make approriate tables reporting possible associations between Hcy and a state or disease and the involved factors mentioned in the text. For example Stroke , hcy levels, outcome or effect on disease evolution, etc
4.Correct English language
Minor comment
Although the authors mention the role of hcy in cardiovascular diseases as a main point in their conclusion this information does not make separate part of their text, thus they should rearrange the beginning of the abstract and the conclusion
Reviewer 2 Answers
· Thank you Sir, we provided to modify the manuscript structure as you suggest;
· Dear sir, thank you for your revision, which helps us and gives us the opportunity to revise the ms, major pathways in which Hcy is involved have been explained;
· Dear sir, we examine in depth the ms, we included table and figure;
· Dear sir, English grammar has been revised by an English native speaker;
· Thank you sir, abstract and conclusions have been revisited.
Reviewer 3 Report
I read the manuscript “The controversial role of Homocysteine in neurological practice”, which was aimed to describe underlying mechanisms involved in
HHcy-induced stress conditions in order to emphasize possible role of HHcy in neurodegenerative disease and stroke.
Common deficiencies in the manuscript prevent initial acceptance for publication.
Many repeated mistakes occurred in the text that in this form is not suitable for publication.
Moreover :
· Circulating levels of Hcy are often reported as >10-2 mol/L. The range of values is confounding and inappropriately reported.
· In the introduction vit B12 has been reported as catalyzer. But this plays a role as cofactor.
· To describe the contribution of methylation reaction in brain synthesis, the term “transimittors” is wrong.
· The protein phosphatase 1 (PPM1) should be mentioned as protein phosphatase methyltransferase 1 (PPM1)
The Authors reported a new possible mechanism of Hcy’s action: its direct activation of the metabotropic receptors.
· No citation has been reported.
From description of results obtained from Curro et al. it would seem that retinoic acid is incubated together with homocysteine. Indeed preliminary RA incubation was carried out to differentiate the neuroblastoma cells.
The authors reported that HHcy produced “an increment of ROS and deactivation of nitric oxide”.
· Indeed, HHcy is responsible for pathogenesis underlying cerebral and other vascular circulatory disorders by impaired synthesis of nitric oxide (NO) in the endothelium or increased production of asymmetric dimethylarginine. A sentence should be added.
Citation 121 and others were no correct.
Author Response
Answers to reviewer’s comments
We would like to thank you for the exhaustive revisions and for the opportunity you have given us.
We have managed with the editorial corrections.
Reviewer 3
I read the manuscript “The controversial role of Homocysteine in neurological practice”, which was aimed to describe underlying mechanisms involved in
HHcy-induced stress conditions in order to emphasize possible role of HHcy in neurodegenerative disease and stroke.
Common deficiencies in the manuscript prevent initial acceptance for publication.
Many repeated mistakes occurred in the text that in this form is not suitable for publication.
Moreover :
· Circulating levels of Hcy are often reported as >10-2 mol/L. The range of values is confounding and inappropriately reported.
· In the introduction vit B12 has been reported as catalyzer. But this plays a role as cofactor.
· To describe the contribution of methylation reaction in brain synthesis, the term “transimittors” is wrong.
· The protein phosphatase 1 (PPM1) should be mentioned as protein phosphatase methyltransferase 1 (PPM1)
The Authors reported a new possible mechanism of Hcy’s action: its direct activation of the metabotropic receptors.
· No citation has been reported.
From description of results obtained from Curro et al. it would seem that retinoic acid is incubated together with homocysteine. Indeed preliminary RA incubation was carried out to differentiate the neuroblastoma cells.
The authors reported that HHcy produced “an increment of ROS and deactivation of nitric oxide”.
· Indeed, HHcy is responsible for pathogenesis underlying cerebral and other vascular circulatory disorders by impaired synthesis of nitric oxide (NO) in the endothelium or increased production of asymmetric dimethylarginine. A sentence should be added.
Citation 121 and others were no correct.
Reviewer 3 Answers
· Dear sir, thank you for your revision, which helps us and gives us the opportunity to revise the ms, Circulating levels of Hcy have been corrected and reported according with previous paper and data from literature;
· Dear sir, the role of vit B12 has been better explained;
· Dear sir, the protein phosphatase 1 (PPM1) has been mentioned as you suggest;
· Dear sir, thank you, citation concerning the new possible mechanism of Hcy’s action (metabotropic receptors) has been reported;
· Dear sir, explanation of results obtained from Curro et al have been reported;
· Dear sir, the correlation of asymmetric dimethylarginine and Hcy, has been reported;
· Citation 121 and other references have been revisited.
Round 2
Reviewer 2 Report
the paper has been substantially improved and the added information is very helpful
Reviewer 3 Report
The revised form have been modified according to suggestions of referee